# Study on an All-Optic Temperature Sensor Based on a Low-Coherent Optical Interferometry

**DOI:** 10.3390/s25247597

**Published:** 2025-12-15

**Authors:** Fanni Gu, Yirui Wen, Sergei Krasovskii, Changsen Sun

**Affiliations:** 1DUT–BSU Joint Institute, Dalian University of Technology, Dalian 116024, China; gfanni@126.com (F.G.); krasovskii.sergei.gen@gmail.com (S.K.); 2College of Optoelectronic Engineering and Instrumentation Science, Dalian University of Technology, Dalian 116024, China; wenyirui@mail.dlut.edu.cn; 3Department of Differential Equations and System Analysis, Faculty of Mechanics and Mathematics, Belarusian State University, 220030 Minsk, Belarus

**Keywords:** spring sensor, intrinsic all-optic temperature sensor, low-coherent Michelson interferometry, thermoelasticity principle

## Abstract

Optical temperature sensors with intrinsic characteristics of explosion-proof are particularly suitable for the petrochemical industry, etc. However, their applications remain limited by environmental compatibility, etc. Here, we developed an all-optic temperature sensor using an anti-bending single-mode optical fiber in a 3.5 m length and a 0.25 mm outer diameter to match a stainless tube with a 0.4 mm inner diameter. The fiber was threaded into the tube, well bonded with epoxy at both ends of the tube, and configured as one arm of a low-coherent Michelson interferometer. Then, the tube with an embedded sensing fiber was fabricated into a spring, whose final length was about 70 mm with an outside diameter of 13 mm. Changes in temperature alter the lengths of the stainless tube spring in a thermoelastic way, thereby modifying the inner fiber length and producing an optical path difference between the sensing fiber and the packaged reference arm of the interferometer. A temperature calibration was carried out from −25 to 65 °C, and the results demonstrated that the hysteresis of the spring sensor was within ±1.16 °C and the sensitivity was 0.34 °C, which was verified by using a platinum resistance temperature sensor (PT-100). This work provides a reference for further intrinsic optical temperature sensor design.

## 1. Introduction

Temperature measurement is one of the most common tasks in our daily life, and many instruments have been developed in a variety of principles [1]. It is still evolving according to the requirements, both in science and technology, such as distributed temperature measurement [2], ultra-high temperature measurement [3], and wide-range temperature measurement [4], etc. Here, we focus on the intrinsic characteristics of explosion-proof methods, which can carry out regular temperature measurements with an all-optic property in order to be used in the petrochemical industry, etc. Some of the all-optic temperature sensors belong to this category and can be available for use, such as fiber Bragg grating (FBG) sensors [5], machine vision [6], fluorescent fiber-optic temperature sensors [7], infrared temperature sensors [8], radiation temperature sensors [9], etc.

The FBG temperature sensor [5] has been widely used in engineering to replace conventional strain gauges. This sensor is an all-optic sensor that requires an interrogator to examine the wavelength shift for temperature resolution. A FBG temperature sensor is also needed to design a module to transfer the temperature to the strain. Thus, how to separate the strain with temperature is somehow a key point for the scheme. A similar thing also happens with Fabry–Perot temperature sensors [10].

The machine vision [6] method can measure temperature distribution in a large area in a remote way. It can determine the temperature field around a ground settlement (GS) sensor. However, the cylindrical structure of the oil tank requires at least three machine vision systems to cover all the surrounding GS sensors, and this leads to installation trouble.

Fluorescent fiber-optic temperature sensors [7] is usually used for high-temperature measurement, even if it is also suitable for liquid temperature measurement.

Infrared [8] and radiation temperature sensors [9] are both non-contact sensors. As a remote method, they are able to measure temperature through an optical fiber, but they still need a separate interrogator.

On the other hand, temperature measurement is usually a topic accompanying with strain measurement in the distributed physical measurement [2], FBG [5], etc. And their cross-sensitivity of temperature with strain is a hot topic among these approaches [11]. It means that temperature and strain are coupled together in these schemes. However, there is no publication to report a temperature measure by using a technique of low-coherent optical interferometer (LCOI), although it has been widely employed for the long-gauge strain measurement. A difficulty in developing a temperature sensor in the principle of LCOI is due to the lack of a high-stability in-fiber mirror to carry out a half-reflection of the light [12,13], by which a specific length of the sensing arm in Michelson configuration can be confined effectively. A similar problem can be found in the distributed situation, in which the width of an optical pulse is employed to control its spatial resolution [14]. One approach was polishing the sensing fiber ends at a right angle and coat them with a thin layer of silver for a reflectivity of 4% to form a partial reflection. This scheme can carry out a quasi-distributed strain sensor in the principle of LCOI. However, it is difficult to configure it as a temperature sensor, due to hysteresis [12].

A GS sensor in the principle of LCOI has been developed for non-uniform GS monitoring of large-storage oil tanks [15]. In that case, the non-uniform temperature field exists surrounding a cylindrical structure of oil tanks, around which the solar irradiation will create a sunlit front and shadow around the tank. In the summer, their temperature differences can exceed over 30 °C [16]. This requires a GS sensor to possess a good temperature performance in order to compare the GS data both from the sunlit front and shadow in an equal accuracy. Thus, to design a temperature sensor in the principle of LCOI becomes necessary in order to implement a corresponding compensation algorithm for the GS sensor. This is also the requirement for fireproofing in the petrochemical industry [16].

This paper proposed an all-optic temperature sensor in the principle of LCOI. Its sensing arm was fabricated into a spring in order to obtain an expected Young’s modulus for temperature sensing in a thermoelastic way. This design can not only firstly implement a temperature sensor in the principle of LCOI but also share one interrogator with the GS sensor. The experimental results proved that this design could obtain an absolute temperature measuring accuracy of ±1.16 °C, and the sensitivity was 0.34 °C with a verification of a platinum resistance temperature sensor (PT-100). Therefore, the temperature and GS measurement can be integrated into one LCOI-based interrogator, and a compact instrumentation can be expected.

## 2. Design of a Spring Temperature Sensor Based on a LCOI

### 2.1. The Principle of a LCOI

A LCOI was configured as given in Figure 1. The system employed a low-coherence broadband light source, a super-luminescent emitting diode (SLED), with a central wavelength at 1310 nm and a full width at half maximum (FWHM) of 45 nm. The coherence length of this source was limited to several tens of micrometers. The interference occurs only when the optical path difference between the reference arm and the sensing arm lies within this coherence length.

The principle of the LCOI given in Figure 1 was that the broadband light emitted from a SLED passed through the first circulator and about half of the energy was reflected by a half-reflection mirror A. Then, it was coupled into the second circulator and projected onto the second half-reflection mirror C and total-reflection mirror D successively, thereby correspondingly forming Path 1 and Path 3. The other half of the energy passed through the half-reflection mirror A, reflected by the total-reflection mirror B, then went the same way to C and D and formed Path 2 and Path 4 one after another. The four paths were summarized in Table 1.

Among them, Path 2 served as a reference arm and Path 3 as a sensing arm, while Path 1 and Path 4 represented the shortest and longest paths among all the four, respectively.

The LCOI is in a Michelson configuration, and the output interference signal reaches its maximum intensity only when the optical path difference (OPD) of the sensing arm, Path 3, along with reference arm, Path 2, tends to zero. This is because the wide spectrum of the SLED bears a short coherent length of approximately 38.14 μm here [15]. The OPD was controlled through the moving stepping motor, and the position of the stepper motor is used to measure the OPD. According to Equation (4) in Ref. [15], this output interference signal detected by PD can be described as(1)$$I\left(Q\right)=I^{\left(1\right)}\left(Q\right)+I^{\left(2\right)}\left(Q\right)+2\sqrt{I^{\left(1\right)}\left(Q\right)}\sqrt{I^{\left(2\right)}\left(Q\right)}\gamma_{12}^{r}(\frac{L_{ABC}-L_{ACD}}{c})$$
where $\gamma_{12}^{r}$ denotes the real part of the complex degree of coherence $\gamma_{12}$; $I^{\left(1\right)}(Q)$ and $I^{\left(2\right)}(Q)$ represent the optical intensity from the two interference paths, respectively; then, the low-coherent interference signal can be detected by the PD and processed in the following data-processing part. $L_{ABC}$ represents the length of Path 2, $L_{ACD}$ is the length of Path 3, and $\Delta L=L_{ABC}-L_{ACD}$ is the OPD between Path 2 and Path 3. In practice, the reference arm, $L_{ABC}$, was packaged in a well temperature-isolated chamber to keep its length stable, so the changes in OPD was determined by the sensing arm, $L_{ACD}$. After being divided by the initial length of the sensing arm $L_{0}$, the strain variations in sensing arm can be expressed as a function of temperature:(2)$$\frac{\Delta L}{L_{0}}=\alpha (T-T_{0})$$
where α is the thermal expansion coefficient of the spring material, stainless here. We determined it by experimental data.

In contrast, the other two OPDs, $L_{AC}$ and $L_{ABCD}$, are far beyond the coherence length of the light source, and thus they cannot interfere.

When temperature is being changed, the stainless-steel spring undergoes a thermal expansion or contraction, and results in a change in the optical length of sensing arm, $L_{ACD}$. The stepping-motor drives the total-reflection mirror B to adjust the length of the reference arm to find where the OPD of Path 2 and Path 3 equals to zero. After which, we recorded the locations of the stepping-motor as a measure of the OPD and compared it with temperature through a calibration procedure. All the regions in Figure 1 apart from the marked red line were insensitive to the results. Thus, this sensor possessed good stability and was quite suitable for the measurement of a quasi-static variable.

### 2.2. The Fabrication of a Spring Sensor

In order to design a temperature sensor with a low hysteresis, the sensing fiber was configured into a spring structure. This can guarantee an all-optic passive temperature sensing. This feature provides an intrinsic safety mechanism and makes the sensor the preferred option for petrochemical industry applications.

The spring sensor was made of a stainless tube threaded with the sensing arm, $L_{ACD}$. The stainless tube had an outside diameter of 1 mm and inner diameter of 0.4 mm; thus, it matched well with the outside diameter, at 0.25 mm, of the sensing arm optical fiber. The fiber was an anti-bending, single-mode optical fiber G657.A2, bearing a minimum bending radius of 7.5 mm. After a 3.5-m-long optical fiber being threaded through a 3-m-long stainless tube bonded with epoxy resin at two ends of the tube, as shown in Figure 2a, a coiling machine was operated to make it into a spring, as shown in Figure 2b, which served as an all-optic temperature sensor. The final size of the spring had a length of about 70 mm and an outside diameter of 13 mm, which satisfies the bending diameter of fiber G657.A2, say 15 mm (7.5 × 2 = 15 mm).

One of the advantages of this spring sensor is its passive all-optic property. And the stainless tube is suitable to measure the temperature of liquid or silicone oil, etc. The strain generated by the spring is less dependent on the deformation in its physical shape, so it is possible to be transformed to match a space where the sensor to be installed, thus keeping its sensitivity determined by thermoelastic effect.

The spring was made of stainless steel for like environmental compatibility. A calculation based on the thermoelastic property of stainless steel had been carried out. The results were discussed in the Conclusion (Section 4).

### 2.3. Data-Processing Part

The data-processing part was mainly to convert the optical interferometric fringe signal into electric information and calibrate the environmental temperature variations into the strain determined by the thermoelastic effect, such as thermal expansion or contraction. During the calibration experiment, the interference fringes generated from the optical interferometer were simultaneously recorded, with the temperature measured by a commercial Pt-100. In this way, two datasets were collected: the central peak of interference fringe displacement (in micrometers) and the corresponding temperature.

The interference fringe displacement was normalized and represented as the strain by dividing it by the sensing fiber length of 3.5 m, and it was then expressed in microstrain (με). By plotting the measured strain as a function of the actual temperature, a linear fitting curve was obtained. The slope of the fitted line represented the sensitivity of the sensor, which provided the conversion coefficient between strain and temperature.

After being calibrated, the spring sensor can be directly applied in the petrochemical industry, such as oil tank GS monitoring. When the spring sensor was placed in the field, only the interferometric displacement needed to be measured. The corresponding temperature was able to be calculated from the established linear relationship, realizing an all-optical detection scheme without any electrical dangers.

## 3. Experiment and Analysis

### 3.1. The Setup for Spring Sensor Calibration

The experimental setup was shown in Figure 3A,B. Due to lack of equipment to cover the whole temperature range from −25 °C to 65 °C, the temperature calibration was separated into two stages: the first was from room temperature (25 °C) to 65 °C, as given in Figure 3A; the second was from room temperature (25 °C) to −25 °C, as given in Figure 3B.

In the first stage, a temperature-controlled water bath (Changzhou Guohua Electric Appliances Co., Ltd., Changzhou, China) was employed to generate a step-like thermostatic varying calibration temperature. The fabricated spring sensor and a PT-100 Platinum resistance thermometer (Zhengzhou Keweilai Electronic Technology Co., Ltd., Zhengzhou, China) were put together in the middle of the bath. The temperatures measured by both PT-100 and the designed spring sensor were simultaneously recorded every 5 min. Then, a strain measured by the spring sensor was calibrated as a temperature sensor.

In the second stage, a low-temperature experiment was carried out using a concrete rapid freeze–thaw testing machine (Model TDR-15D, with a power supply of 380 $\tilde{V}$ and 50 Hz, Wuhu Huance Instrument Equipment Co., Ltd., Wuhu, China). The system employed an antifreeze coolant circulation to decrease the temperature to approximately −25 °C. The spring sensor was immersed into the antifreeze coolant together with the PT100 thermometer as the insert of Figure 3B. After the temperature reached each setpoint, it was maintained to ensure thermal equilibrium. When the temperature of −25 °C was reached, the heating power was turned off, and the system was allowed to cool naturally down to room temperature. Three circulations from room temperature to −25 °C were set with a step of 5 °C for both increasing and decreasing processes. Total three roundtrip calibration curves were obtained within three days, and a fitting curve was achieved, as presented in Figure 4.

The vertical dashed line represented the room temperature, by which the range of the calibration temperature was separated into the two stages, and the strain from the low-coherent interferometer of each process was normalized as zero and connected to form Figure 4. The right side of the dashed line corresponded to the first stage given by Figure 3A from room temperature to 65 °C; and the left side corresponded to the second stage given by Figure 3B from room temperature to low temperature −25 °C. The insert figure indicated the hysteresis of the strain between the increasing and decreasing temperatures as well as the point of maximum error.

As shown in Figure 4, the strain measured by the spring sensor corresponded linearly to the bath temperature. This indicated that the designed spring sensor was linear within the calibrated temperature range. The fitting results showed a well-linear relationship with a sensitivity of 14.78 με/℃, which was calculated as the slope of the fitting curve in Figure 4 and is to be discussed in Section 4. The error bars showed the deviation of the measured strain values, which were plotted with ten-degree intervals. The error was mainly from the mechanical hysteresis of the spring sensor. In addition, the following factors also contributed to the error: (1) internal friction within the stainless-steel spring tube material during thermal expansion/contraction cycles; (2) potential micro-slip at the fiber-tube interface along the unbonded sections due to differential thermal expansion between the silica fiber and stainless-steel tube; (3) transient strain mismatches caused by differing response times of the constituent materials.

### 3.2. Results Comparison and Discussion

The results were fitted separately for the increasing and decreasing processes. The results demonstrated that both increase and decrease in temperature exhibited a good linear relationship between the strain and temperature. This proved the design of an all-optic temperature sensor in a spring structure. The maximum hysteresis error, approximately 2.31 °C (±1.16 °C) between the temperature increases and decreases, happened at 43.7 °C, marked as a red error bar in the inserted figure given in Figure 4. This hysteresis corresponded to about 2.6%, which was calculated as 2.31/90≈2.6% in the full scale of the calibrated temperature range of 90 °C, which was calculated as 65 °C−(−25 °C)=90 °C. This was able to meet most of the cases required in engineering temperature measurement.

Even if it was not an alternative to FBG, but rather a complementary technology, compared with the conventional FBG sensors, the designed spring sensor had the following advantages:It was an all-optic intrinsic temperature sensor bearing a good hysteresis property.Its intrinsic property made it get rid of the tendency for grating performance deterioration inherent in a FBG sensor.It allowed a larger dynamic range than the FBG sensor due to the single-mode fiber being able to approach about 5000 με. It was supposed to be serving as a long-term measurement according to the performance of optical fiber.From an engineering point of view, this spring sensor was able to deform itself to adapt to the application scene. This property made it highly suitable for practical applications, especially in cases where a sensor limitation was required.In our cases, it was well matched to a successful GS sensor, which was also configured in a low-coherence interferometry. Thus, the GS of an oil tank could be measured with an accompanying temperature simultaneously by using one integrated interrogator.

## 4. Discussion and Conclusions

An intrinsic spring temperature sensor was designed and fabricated. The calibration results demonstrated its effectiveness. The sensitivity based on the calibrated curve given in Figure 4 was 14.78 με/°C, from which the temperature sensitivity was able to be estimated as 0.34 °C. This result was determined by the resolution of the interrogator, which was 5 με in our experiment and was determined by the accuracy of the stepping motor given in Figure 1.

According to the theory of thermoelasticity, the Duhamel–Neumann relation [17] is known as(3)$$\varepsilon_{ij}=S_{ijkl}\sigma_{kl}+\alpha_{ij}(T-T_{0})$$
where $\varepsilon_{ij}$ is the strain tensor, $S_{ijkl}$ is the compliance tensor (inverse of the stiffness tensor $C_{ijkl}$), $\sigma_{kl}$ is the stress tensor, $\alpha_{ij}$ is the thermal expansion tensor, and $T-T_{0}$ is the temperature change.

For Equation (3), the equivalent form [18] can be derived as(4)$$\varepsilon_{ij}=\frac{1+\nu }{E}\sigma_{ij}-\frac{\nu }{E}\delta_{ij}\sigma_{kk}+\alpha (T-T_{0})\delta_{ij}$$
where E is Young’s modulus of the material, ν is Poisson’s ratio, α is the thermal expansion coefficient, $\delta_{ij}$ are Kronecker deltas.

When thermally exposed to a free stainless tube that has no external mechanical limitations, stresses do not occur in the material, but a linear strain is observed, proportional to the temperature change. This effect can be described by the thermoelasticity equation in the form of $\Delta L/L=\;\alpha (T-T_{0})$, where α is the coefficient of linear thermal expansion of the stainless tube, and L is the initial tube length (L = 3.5 m at $T_{0}=28.25\;{}^{\circ}C$ in the experiment with heating, and $T_{0}=25.0\;{}^{\circ}C$ in the experiment with cooling). In the absence of fasteners, the tube deforms freely, which eliminates the appearance of internal stresses, despite the presence of thermal deformation.

The maximum hysteresis error was 2.31 °C(±1.16 °C) between the temperature increases and decreases.

For the increasing process, ε=ΔL/L=15.05×(T−25.0), where $T_{0}=25.0$ °C, α=15.052 µm/(m·°C).

For the decreasing process, ε=14.65×(T−28.25), where $T_{0}$ = 28.25 °C, α = 14.65 µm/(m·°C). Thus, the fitting of calibration curve turns out as ε=14.78×(T−28.22), where $T_{0}$ = 28.22 °C and α = 14.78 µm/(m·°C) are the fitting parameters.

Of course, there are nonlinear effects, since thermoelastic expansion must lead to friction, local clamping, or even micro-sliding between the fiber and the inner surface of the tube. Additionally, the result of full thermoelastic derivation depends on the spring geometry (wire diameter, coil radius, number of turns, etc.). It also depends on the mechanical transfer factor between tube expansion and inner fiber strain, effects of coil deformation, axial stiffness, or friction between the fiber and inner tube. However, our experiments have shown that both heating and cooling nonlinear effects lead to deviations of no more than 0.19% of the measured optical path length from the calculated linear approximation. This is sufficient accuracy for our purposes.

This work presented a design and fabrication of an all-optic temperature spring sensor. The experimental test demonstrated that the proposed spring sensor could carry out temperature measurement and provided a possibility to carry out a GS of oil tank measurement with fine temperature compensation. This provided a reference for future intrinsic sensor development.

For further optimization, we will focus on two key aspects:

The first is to improve temperature resolution. We will systematically modify the spring structure parameters (e.g., diameter, pitch, and number of coils) through iterative testing to identify the optimal configuration that maximizes strain output per unit temperature change.

The second is to reduce hysteresis. We will implement a continuous or multi-point bonding technique along the fiber-tube interface to minimize micro-slip and ensure more uniform strain transfer.

The third is choosing an optical fiber with a low stress birefringence. Even if an optical fiber can be a good strain material by acting as a basis of FBG, for example, when it is being coiled along with the spring, a low-stress-birefringence fiber should be preferred. In the future, we can separate this effect by using a specifically designed optical scheme. A long-term field validation of both its durability and reliability is also an on-going work.

## Figures and Tables

**Figure 1 sensors-25-07597-f001:**
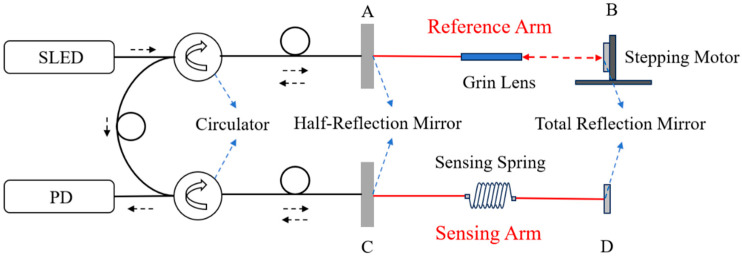
The configuration of the all-optic temperature-sensing system based on LCOI included the sensing spring, optical configuration, and signal detection part. SLED: super-luminescent emitting diode; PD: photodetector; A and C represented half-reflection mirror; B and D denoted total reflection mirror.

**Figure 2 sensors-25-07597-f002:**
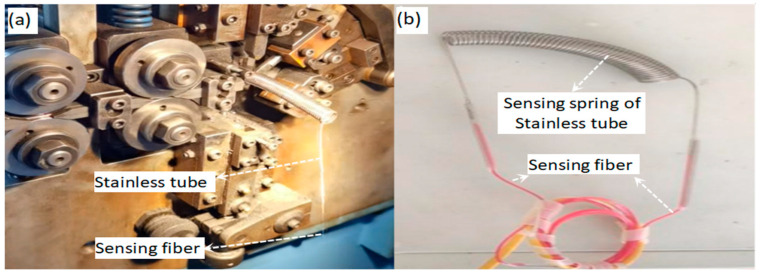
The fabrication of the spring temperature sensor. (**a**) The coiling machine was employed to fabricate a stainless tube spring threaded with a sensing fiber; (**b**) the fabricated sensing spring sensor.

**Figure 3 sensors-25-07597-f003:**
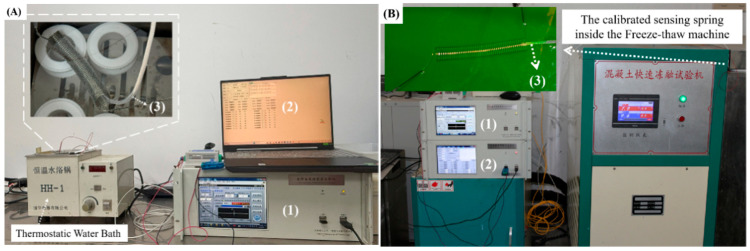
The setup for spring sensor calibration. (**A**) Calibration test from room temperature to 65 °C; (**B**) test from room temperature to −25 °C. Both of them shared the same instruments, including (1) the low-coherence interferometer; (2) the PT100 data-acquisition unit; (3) the PT100 temperature sensor.

**Figure 4 sensors-25-07597-f004:**
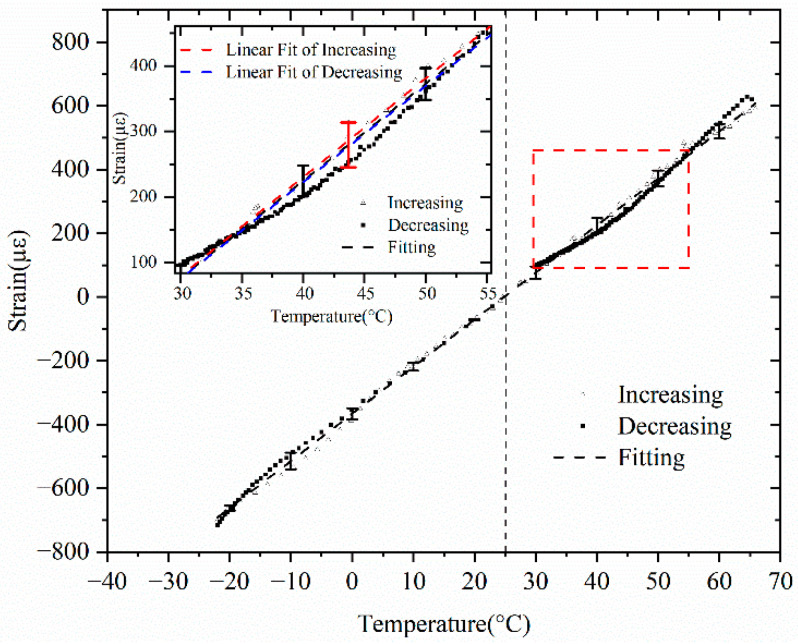
Calibration curve of the designed spring temperature sensor by using a PT100 resistance thermometer. The data underlying this curve can be found in Appendix A “Data of Figure 4”.

**Table 1 sensors-25-07597-t001:** Summary of the four paths and their optical meanings.

Path	Transmission Path	Optical Meaning
Path1	A+C	Shortest path
Path2	A+B+C	Reference arm
Path3	A+C+D	Sensing arm
Path4	A+B+C+D	Longest path

## Data Availability

The data generated or analyzed as part of the research are not publicly available. This research keeps going on and the data will be disclosed with permission of the oil tank running corporation.

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
