# Peer review of "Study on an All-Optic Temperature Sensor Based on a Low-Coherent Optical Interferometry"

_sensors, 2025, doi:10.3390/s25247597_

Round 1

Reviewer 1 Report

Comments and Suggestions for Authors

See the attachment.

Comments on the Quality of English Language

The manuscript contains numerous grammatical issues, long sentences, and repeated descriptions. A professional language edit is required.

Author Response

Thanks for the reviewers’ comments. We answered the reviewers’ questions one by one in the document below.

Reviewer 2 Report

Comments and Suggestions for Authors

The paper presents an interesting approach to developing an all-optic temperature sensor. The paper, "Study on an all-optic temperature sensor based on a low-coherent optical interferometry," is interesting and worth disseminating. However, several aspects of the methodology, results presentation, and discussion require significant refinement to meet the standards of a technical publication. A major revision is recommended, with particular attention to the following points:

  1. The current diagram is quite complex as well as blurred image. A clearer, simplified schematic of the Michelson interferometer specifically highlighting the reference and sensing arms, and how the spring sensor integrates, would greatly improve understanding. Labeling of components could be more distinct.
  2. The explanation of four optical paths is desired, a visual representation or a table summarizing the components involved in each path would be beneficial for clarity, especially for Paths 2 and 3, which are crucial for interference.
  3. The text mentions "interference occurs only when the optical path difference... lies within this coherence length," but a more explicit explanation of how this is achieved and maintained for Path 2 and Path 3 (as stated in line 125) would strengthen the foundational understanding.
  4. Eq. (1) is introduced without clear context or derivation within the paper, referencing Ref. [6]. For a self-contained technical publication, a brief explanation of what the variables represent and why this specific equation is applicable to the sensor's operation is needed.
  5. Fig. 2a shows a coiling machine, a more detailed description of its operation and the specific parameters (e.g., coiling tension, mandrel diameter, number of turns) used to fabricate the spring sensor would be valuable for reproducibility.
  6. The images in Fig. 3 (a) and (b) are somewhat blurry and difficult to discern the details of the experimental setup. Higher-resolution images with clearer labels for each component (e.g., PT-100, spring sensor, freeze-thaw machine) would significantly enhance this section.
  7. This work states the calibration was separated into two stages (room temp to 65°C and room temp to -25°C) due to "lacking of equipment to cover the whole range." Elaborate on why a single setup couldn't cover the full range and if this separation introduces any potential discrepancies in the combined results.
  8. The discussion of hysteresis in Fig. 4 mentions a "red error bar" at 43.7°C. It would be helpful to visually highlight this specific error bar in the figure for easier identification. Further analysis of the sources of this hysteresis beyond "mechanical hysteresis" would also be beneficial.
  9. Authors should list the advantages of this work over FBG sensors, a more quantitative comparison, perhaps in a table, detailing specific performance metrics (e.g., sensitivity, resolution, operating range, cost implications) would provide a stronger argument for the proposed sensor's benefits.
  10. The conclusion mentions future optimization, including improving temperature resolution and reducing hysteresis. Providing specific technical approaches or design modifications to be considered for these improvements would add depth to the future work section.
  11. The authors should thoroughly study other work and make a strong case in the introduction. There should be a minimum of more than 30 references quoted to demonstrate a comprehensive understanding of the current state of the art and the novelty of the presented work.

Author Response

(The authors gave the same response as above.)

Reviewer 3 Report

Comments and Suggestions for Authors

This manuscript reports a temperature sensor based on low-coherence optical interferometry, presenting a sensitivity of 0.34 °C over a temperature range from –25 °C to 65 °C. After a detailed review of the paper, I find that the measurement concept of the proposed sensor is not sufficiently established and the experimental results are rather limited. For the reasons described below, I am unable to recommend this manuscript for publication.

1. The authors claim advantages over other sensing technologies such as FBG sensors introduced in the Introduction; however, no superior performance has been clearly demonstrated. In fact, the OLCR system—which requires mechanical scanning—appears to be less suitable for practical sensor applications. The authors need to articulate the technical advantages of the proposed approach compared with existing sensor technologies.

2. The term all optical is not appropriate in the title. Most fiber-optic sensors are inherently all-optical, and this may mislead readers into expecting a special or unique technology. Although the authors provide some theoretical description of OLCR, the explanation is insufficient. A detailed description of how temperature variations are actually detected—e.g., how the measured signal changes with temperature—is missing.

3. It appears that the OLCR module used in the experiment is a commercial system. In such a case, a detailed description and characterization of the sensor head are essential, supported by experiments. However, based on Section 2.2, the structural description and the provided photograph are not sufficient to understand the sensor head clearly. Moreover, given the 3.5 m-long fiber, the step-motor scanning range for OLCR should exceed at least 1.5 m, yet no discussion of measurement time is provided. A clear conceptual diagram of the sensor structure and its characteristics is required.

4. The sensor seems to estimate temperature by measuring the strain of a spring. For a 3.5 m fiber, temperature-induced refractive-index changes will affect the OLCR signal, but no analysis is presented. In addition, the spring is easily affected by gravity or external forces introduced during installation. These potential influences should be discussed.

5. In Fig. 4 (page 6), the authors attribute the error in the 40–60 °C range to “mechanical hysteresis of the spring sensor,” but there is no explanation of why this occurs or how it can be reduced. Linearity is a key performance parameter of sensors, and sufficient discussion is required.

6. The authors argue that the proposed sensor exhibits a larger dynamic range than FBG temperature sensors; however, FBG sensors typically maintain linear response from –20 °C up to around 200 °C. Since the proposed method detects strain while FBG measures wavelength shift directly due to temperature, comparing dynamic range is not appropriate without a clear technical basis.

In summary, substantial additional experimental work and major revisions are required before the manuscript can be considered for publication.

Author Response

(The authors gave the same response as above.)

Round 2

Reviewer 1 Report

Comments and Suggestions for Authors

The revised manuscript is acceptable

Reviewer 2 Report

Comments and Suggestions for Authors

The authors have made substantial revisions in response to your comments, significantly improving the manuscript's rigor, clarity, and comprehensiveness. They have provided quantitative data, detailed explanations, and revised figures where requested.

I would recommend accepting the revised manuscript.

Reviewer 3 Report

Comments and Suggestions for Authors

The authors have properly addressed the comments and revised the manuscript accordingly. Based on the satisfactory responses and revisions, I agree with the publication of this manuscript.